# Learning Discrete Weights Using the Local Reparameterization Trick

**Oran Shayer**
General Motors Advanced Technical Center - Israel
Department of Electrical Engineering, Technion
oran.sh@gmail.com

**Dan Levi**
General Motors Advanced Technical Center - Israel
dan.levi@gm.com

**Ethan Fetaya**
University of Toronto
Vector Institute
ethanf@cs.toronto.edu

## Abstract

Recent breakthroughs in computer vision make use of large deep neural networks, utilizing the substantial speedup offered by GPUs. For applications running on limited hardware, however, high precision real-time processing can still be a challenge. One approach to solving this problem is training networks with binary or ternary weights, thus removing the need to calculate multiplications and significantly reducing memory size. In this work, we introduce LR-nets (Local reparameterization networks), a new method for training neural networks with discrete weights using stochastic parameters. We show how a simple modification to the local reparameterization trick, previously used to train Gaussian distributed weights, enables the training of discrete weights. Using the proposed training we test both binary and ternary models on MNIST, CIFAR-10 and ImageNet benchmarks and reach state-of-the-art results on most experiments.

## 1 Introduction

Deep Neural Networks have been the main driving force behind recent advancement in machine learning, notably in computer vision applications. While deep learning has become the standard approach for many tasks, performing inference on low power and constrained memory hardware is still challenging. This is especially challenging in autonomous driving in electric vehicles where high precision and high throughput constraints are added on top of the low power requirements.

One approach for tackling this challenge is by training networks with binary $\{\pm 1\}$ or ternary $\{-1, 0, 1\}$ weights (Courbariaux et al., 2015; Rastegari et al., 2016) that require an order of magnitude less memory and no multiplications, leading to significantly faster inference on dedicated hardware. The problem arises when trying to backpropagate errors as the weights are discrete. One heuristic suggested in Courbariaux et al. (2015) is to use stochastic weights $w$, sample binary weights $w_b$ according to $w$ for the forward pass and gradient computation and then update the stochastic weights $w$ instead. Another idea, used by Hubara et al. (2016) and Rastegari et al. (2016), is to apply a "straight-through" estimator $\frac{\partial \text{sign}}{\partial r} = r\mathbb{1}[|r| \leq 1]$. While these ideas were able to produce good results, even on reasonably large networks such as ResNet-18 (He et al., 2016), there is still a large gap in prediction accuracy between the full-precision network and the discrete networks.

In this paper, we attempt to train neural networks with discrete weights using a more principled approach. Instead of trying to find a good "derivative" to a non-continuous function, we show how we can find a good smooth approximation and use its derivative to train the network. This is based on the simple observation that if at layer $l$ we have stochastic (independent) weights $w_{ij}^l$, then the pre-activations $z_i^l = \sum_j w_{ij}^l h_j^l$ are approximately Gaussian according to the (Lyapunov) central limit theorem (CLT). This allows us to model the pre-activation using a smooth distribution and use the reparameterization trick (Kingma & Welling, 2014) to compute derivatives. The idea of mod-

eling the distribution of pre-activation, instead of the distribution of weights, was used in Kingma et al. (2015) for Gaussian weight distributions where it was called the local reparameterization trick. We show here that with small modifications it can be used to train discrete weights and not just continuous Gaussian distributions.

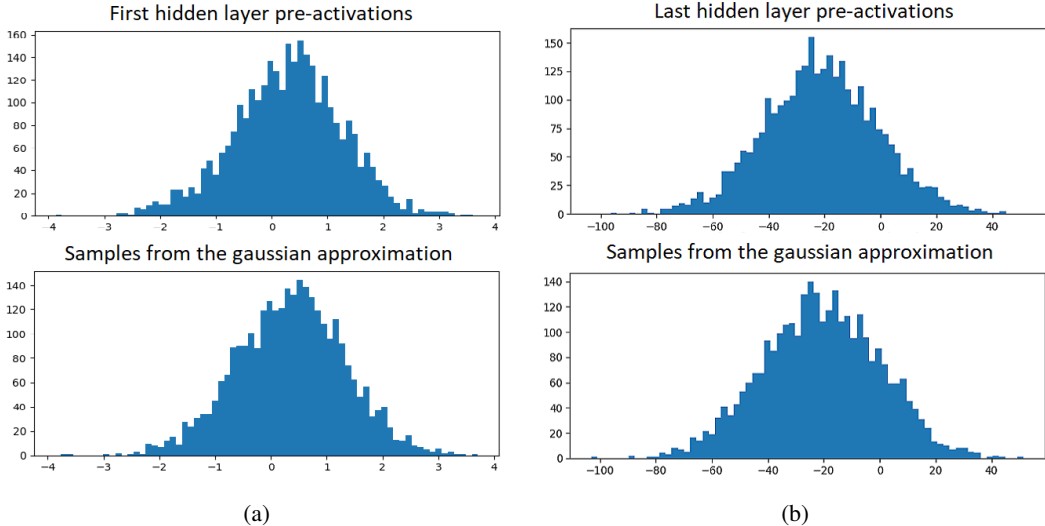

Figure 1: The top histogram in each subfigure shows the pre-activation of a random neuron, $z_i^l$, which is calculated in a regular feed-forward setting when explicitly sampling the weights. The bottom shows samples from the approximated pre-activation using Lyapunov CLT. (a) refers to the first hidden layer whereas (b) refers to the last. We can see the approximation is very close to the actual pre-activation when sampling weights and performing standard feed-forward. In addition, we see it even holds for the first hidden layer, where the number of elements is not large (in this example, 27 elements for a $3 \times 3 \times 3$ convolution).

We experimented with both binary and ternary weights, and while the results on some datasets are quite similar, ternary weights were considerably easier to train. From a modeling perspective restricting weights to binary values $\{\pm 1\}$ forces each neuron to affect all neurons in the subsequent layer making it hard to learn representations that need to capture several independent features.

In this work, we present a novel and simple method for training neural networks with discrete weights. We show experimentally that we can train binary and ternary networks to achieve state-of-the-art results on several datasets, including ResNet-18 on ImageNet, compared to previously proposed binary or ternary training algorithms. On MNIST and CIFAR-10 we can also almost match the performance of the original full-precision network using discrete weights.

## 2 RELATED WORK

The closest work to ours, conceptually, is expectation-backpropagation (Soudry et al., 2014) in which the authors use the CLT approximation to train a Bayesian mean-field posterior. On a practical level, however, our training algorithm is entirely different: their forward pass is deterministic while ours is stochastic. In addition, we show results on much larger networks. Another close line of work is Wang & Manning (2013) where the authors used the CLT to approximate the dropout noise and speed-up training, but do not try to learn discrete weights.

Recently there is great interest in optimizing discrete distributions by using continuous approximations such as the Gumbel-softmax (Maddison et al., 2017; Jang et al., 2017), combined with the reparameterization trick. However, since we are looking at the pre-activation distribution, we use the simpler Gaussian approximation as can be seen in Fig 1. In appendix A we compare our method to the Gumbel-softmax relaxation and show that our method leads to much better training.

One approach for discrete weight training suggested in Courbariaux et al. (2015) is to use stochastic weights, sample binary weights and use it for the forward pass and gradient computation, computing gradients as if it was a deterministic full precision network, and then update the stochastic weights. The current leading works on training binary or ternary networks, Rastegari et al. (2016); Li et al. (2016); Hubara et al. (2016), are based on a different approach. They discretize during the forward pass, and back-propagate through this non-continuous discretization using the "straight-through" estimator for the gradient. We note that Rastegari et al. (2016); Hubara et al. (2016); Soudry et al. (2014) binarize the pre-activation as well.

Another similar work based on the "straight-through" estimator is Zhu et al. (2017). The authors proposed training a ternary quantization network, where the discrete weights are not fixed to $\{-1, 0, 1\}$ but are learned. This method achieves very good performance, and can save memory compared to a full precision network. However, the use of learned discrete weights requires using multiplications which adds computational overhead on dedicated hardware compared to binary or ternary networks.

An alternative approach to reducing memory requirements and speeding up deep networks utilized weight compression. Louizos et al. (2014); Han et al. (2016) showed that deep networks can be highly compressed without significant loss of precision. However, the speedup capabilities of binary or ternary weight networks, especially on dedicated hardware, is much higher.

## 3  OUR METHOD

We will now describe in detail our algorithm, starting with the necessary background. We use a stochastic network model in which each weight $w_{ij}^l$ is sampled independently from a multinomial distribution $\mathcal{W}_{ij}^l$ . We use $W$ to mark the set of all parameters $w_{ij}^l$ and $\mathcal{W}$ to mark the distribution over $W$. Given a loss function $\ell$ and a neural network $f$ parametrized by $W$ our goal is to minimize

$$L(\mathcal{W}) = \mathbb{E}_{W \sim \mathcal{W}} \left[ \sum_{i=1}^{N} \ell \left( f(x_i, W), y_i \right) \right] \tag{1}$$

### 3.1  BACKGROUND

The standard approach for minimizing $L(\mathcal{W})$ with discrete distributions is by using the log-derivative trick (Williams, 1992):

$$\nabla L(\mathcal{W}) = \mathbb{E}_{W \sim \mathcal{W}} \left[ \sum_{i=1}^{N} \ell \left( f(x_i, W), y_i \right) \nabla \log(P(W)) \right] \tag{2}$$

While this allows us to get an unbiased estimation of the gradient, it suffers from high variance, which limits the effectiveness of this method.

For **continuous** distributions Kingma & Welling (2014) suggested the *reparameterization trick* - instead of optimizing $\mathbb{E}_{p(x)}[f(x)]$ for $p(x)$ we parametrize $x = g(\epsilon; \theta)$ where $\epsilon$ is drawn from a known fixed distribution $p(\epsilon)$ (usually Gaussian) and optimize $\mathbb{E}_{p(\epsilon)}[f(g(\epsilon, \theta))]$ for $\theta$. We can sample $\epsilon_1, ..., \epsilon_m$ and use the Monte-Carlo approximation:

$$\nabla_\theta \mathbb{E}_{p(\epsilon)}[f(g(\epsilon, \theta))] \approx \sum_{i=1}^{m} \nabla_\theta f(g(\epsilon_i, \theta)) \tag{3}$$

In further work by Kingma et al. (2015), it was noticed that if we are trying to learn Bayesian networks, sampling weights and running the model with different weights is quite inefficient on GPUs. They observed that if the weight matrix[1] $W$ is sampled from independent Gaussians $W_{ij} \sim \mathcal{N}(\mu_{ij}, \sigma_{ij}^2)$, then the pre-activations $z = Wh$ are distributed according to

$$z_i \sim \mathcal{N}(\sum_j \mu_{ij} h_j, \sum_j \sigma_{ij}^2 h_j^2) \tag{4}$$

This allows us to sample pre-activations instead of weights, which they called the local reparameterization trick, reducing run time considerably.

---

[1]This is done per-layer. To simplify notation we ignore the layer index $l$

### 3.2 DISCRETE LOCAL REPARAMETERIZATION

The work by Kingma et al. (2015) focused on Gaussian weight distributions, but we will show how the local reparameterization allows us to optimize networks with discrete weights as well. Our main observation is that while eq. 4 holds true for Gaussians, from the (Lyapunov) central limit theorem we get that $z_i = \sum_j w_{ij} h_j$ should still be approximated well by the same Gaussian distribution $z_i \sim \mathcal{N}(\sum_j \mu_{ij} h_j, \sum_j \sigma_{ij}^2 h_j^2)$. The main difference is that $\mu_{ij}$ and $\sigma_{ij}^2$ are the mean and variance of a multinomial distribution, not a Gaussian one. Once the discrete distribution has been approximated by a smooth one, we can compute the gradient of this smooth approximation and update accordingly.

This leads to a simple algorithm for training networks with discrete weights - let $\theta_{ij}$ be the parameters of the multinomial distribution over $w_{ij}$. At the forward pass, given input vector $h$, we first compute the weights means $\mu_{ij} = \mathbb{E}_{\theta_{ij}}[w_{ij}]$ and variances $\sigma_{ij}^2 = \text{Var}_{\theta_{ij}}[w_{ij}]$. We then compute the mean and variance of $z_i$, $m_i = \mathbb{E}[z_i] = \sum_j \mu_{ij} h_j$ and $v_i^2 = \text{Var}[z_i] = \sum_j \sigma_{ij}^2 h_j^2$. Finally we sample $\epsilon \sim \mathcal{N}(0, \mathbf{I})$ and return $\hat{z} = m + v \odot \epsilon$, where $\odot$ stands for the element-wise multiplication. We summarize this in Algorithm 1.

During the backwards phase, given $\frac{\partial L}{\partial \hat{y}}$ we can compute

$$\frac{\partial L}{\partial \theta_{ij}} = \frac{\partial L}{\partial \hat{z}_j} \cdot \frac{\partial \hat{z}_j}{\partial \theta_{ij}} = \frac{\partial L}{\partial \hat{z}_j} \left( \frac{\partial m_j}{\partial \theta_{ij}} + \epsilon_j \frac{\partial v_j}{\partial \theta_{ij}} \right) = \frac{\partial L}{\partial \hat{z}_i} \left( h_j \frac{\partial \mu_{ij}}{\partial \theta_{ij}} + \frac{\epsilon_j h_j^2}{2v_j} \frac{\partial \sigma_{ij}^2}{\partial \theta_{ij}} \right) \quad (5)$$

and similarly compute $\frac{\partial L}{\partial x_i}$ to backpropagate. We can then optimize using any first order optimization method. In our experiments we used Adam (Kingma & Ba, 2015).

---

**Algorithm 1** Discrete layer forward pass

**INPUT:** Vector $h \in \mathbb{R}^{d\_in}$
**PARAMETERS:** Multinomial parameters $\theta_{ij}$ for each weight.
 1: Compute $\mu_{ij} = \mathbb{E}_{\theta_{ij}}[w_{ij}]$ and $\sigma_{ij}^2 = \text{Var}_{\theta_{ij}}[w_{ij}]$
 2: Compute $m_i = \sum_j \mu_{ij} h_j$ and $v_i^2 = \sum_j \sigma_{ij}^2 h_j^2$
 3: Sample $\epsilon \sim \mathcal{N}(0, I)$
**RETURN:** $m + v \odot \epsilon$

---

## 4 IMPLEMENTATION DETAILS

In section 3 we presented our main algorithmic approach. In this section, we discuss the finer implementation details needed to achieve state-of-the-art performance. We present the numerical experiments in section 5.

For convenience, we represent our ternary weights using two parameters, $a_{ij}^l$ and $b_{ij}^l$, such that $p(w_{ij}^l = 0) = \sigma(a_{ij}^l)$ and $p(w_{ij}^l = 1|w_{ij}^l \neq 0) = \sigma(b_{ij}^l)$. For binary networks we simply fix $p(w_{ij}^l = 0) = 0$.

### 4.1 INITIALIZATION

In a regular DNN setting, we usually either initialize the weights with a random initializer, e.g Xavier initializer (Glorot & Bengio, 2010) or set the initial values of the weights to some pretrained value from a classification network or another similar task. Here we are interested in initializing distributions over discrete weights from pretrained continuous deterministic weights, which we shall denote as $\widetilde{W}$. We normalized the pretrained weights $\widetilde{W}$ beforehand to be mostly in the $[-1, 1]$ range, by dividing the weights in each layer by $\sigma^l$, the standard deviation of the weights in layer $l$.

We first explain our initialization in the ternary setting. Our aim is threefold: first, we aim to have our mean value as close as possible to the original full precision weight. Second, we want to have low variance. Last, we do not want our initial distributions to be too deterministic, in order not to start at a bad local minima.

If we just try to minimize the variance while keeping the mean at $\tilde{w}_{ij}^l$ then the optimal solution would have $p(w_{ij}^l = 0) = 1 - |\tilde{w}_{ij}^l|$ and the rest of the probability is at $+1$ or $-1$, depending on the sign of $\tilde{w}_{ij}^l$. This is not desirable, as our initialized distribution in such case could be too deterministic and always assign zero probability to one of the weights. We therefore modify it and use the following initialization:

We initialize the probability of $p(w_{ij}^l = 0)$ to be

$$p(w_{ij}^l = 0) = p_{max} - (p_{max} - p_{min}) \cdot |\widetilde{w}_{ij}^l| \tag{6}$$

where $p_{min}$ and $p_{max}$ are hyperparameters (set to 0.05 and 0.95 respectively in our experiments). Using the initialization proposed in eq. 6, $p(w_{ij}^l = 0)$ gets the maximum value $p_{max}$ when $\widetilde{w}_{ij}^l = 0$, and decays linearly to $p_{min}$ when $|\widetilde{w}_{ij}^l| = 1$ . Next, we set $p(w_{ij}^l = 1 \mid w_{ij}^l \neq 0)$ so that the initialized mean of the discrete weight is equal to the original full precision weight:

$$\mathbb{E}[w_{ij}^l] = [2 \cdot p(w_{ij}^l = 1 \mid w_{ij}^l \neq 0) - 1] \cdot (1 - p(w_{ij}^l = 0)) \tag{7}$$

In order for the expression in eq. 7 to be equal to $\widetilde{w}_{ij}^l$, we need to set:

$$p(w_{ij}^l = 1 \mid w_{ij}^l \neq 0) = 0.5 \cdot \left(1 + \frac{\widetilde{w}_{ij}^l}{1 - p(w_{ij}^l = 0)}\right) \tag{8}$$

Since not all the weights $\widetilde{W}$ are in the range of $[-1, 1]$, values from equations 6, 8 can be larger than one or negative. Hence, we clip these values to be in the range $[p_{min}, p_{max}]$.

For a binary setting, we simply set $p(w_{ij}^l = 0) = 0$.

## 4.2 INCREASING ENTROPY

During some of our experiments with ternary networks, we found that many weight distributions converged to a deterministic value, and this led to suboptimal performance. This is problematic for two main reasons: First, this could lead to vanishing gradients due to saturation. Second, due to the small number of effective random variables, our main assumption - that the pre-activation distribution can be approximated well by a Gaussian distribution - is violated. An example of such case can be seen in fig. 2a.

This can be solved easily by adding $L_2$ regularization on the distribution parameters (before sigmoid, $a_{ij}^l$ and $b_{ij}^l$) to penalize very low entropy distributions. We emphasize that this regularization term was used mainly to help during training, not to reduce over-fitting. From here on, we shall denote this regularization hyper-parameter as *probability decay*, to distinguish it from the weight decay used on the last full-precision layer. We note that in practice, only an extremely small probability decay is needed, but it had a surprisingly large effect on the overall performance. An example of such a case can be seen in fig. 2b. A closer examination of the average entropy throughout the training and the effect of the proposed regularization, can be found in appendix B.

## 4.3 REDUCING ENTROPY

In the binary setting, we experienced the opposite in some experiments. Since we do not have a zero weight, we found that many of the weight probabilities learned were stuck at mid-values around 0.5 so that the expected value of that weight would remain around zero. That is not a desirable property, since we aim to get probabilities closer to 0 or 1 when sampling the final weights for evaluation. For this case, we add a beta density regularizer on our probabilities, $R(p) = p^{\alpha-1}(1-p)^{\beta-1}$ with $\alpha, \beta = 2$. From here on we shall denote this regularization hyper-parameter as *beta parameter*. It is important to note that we are using $R(p)$ as our regularizer and not $-\log(R(p))$ (with $\alpha, \beta < 1$) as might be more standard. This is because $-\log(R(p))$ has a much stronger pull towards zero entropy. In general we note that unlike ternary weights, binary weights require more careful fine-tuning in order to perform well.

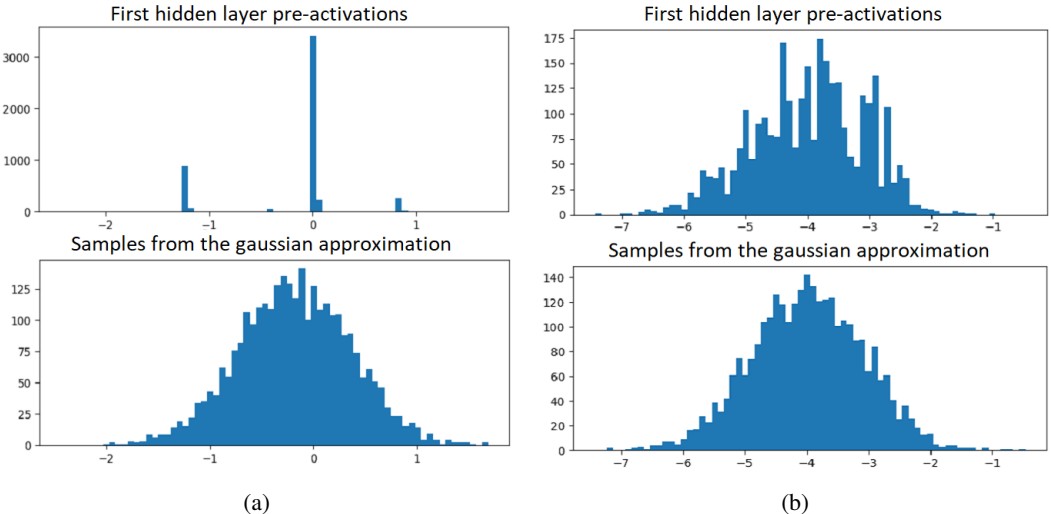

Figure 2: (a) is an example of a neuron from the first hidden layer at the end of the training, without a probability decay term. Since weight distributions converged to deterministic values, randomness is very small, causing the CLT approximation to no longer hold. (b) is an example of a neuron from the same layer at the end of the training, but from an experiment with probability decay. In this experiment our approximation holds much better.

## 4.4 EVALUATION

For evaluation, we sample discrete weights according to our trained probabilities and perform the standard inference with the sampled weights. We can sample the weights multiple times and pick those that performed best on the validation set. We note that since our trained distributions have low entropy, the test accuracy does not change significantly between samples, leading to only a minor improvement in overall performance.

## 5 EXPERIMENTS

We conducted extensive experiments on the MNIST, CIFAR-10 and ImageNet (ILSVRC2012) benchmarks. We present results in both binary and ternary settings. In the binary setting, we compare our results with BinaryConnect (Courbariaux et al., 2015), BNN (Hubara et al., 2016), BWN and XNOR-Net (Rastegari et al., 2016). In the ternary setting, we compare our results with ternary weight networks (Li et al., 2016).

In previous works with binary or ternary networks, the first and last layer weights remained in full precision. In our experiments, the first layer is binary (or ternary) as well (with an exception of the binary ResNet-18), leading to even larger memory and energy savings. As previous works we keep the final layer in full precision as well.

## 5.1 MNIST

MNIST is an image classification benchmark dataset, containing 60K training images and 10K test images from 10 classes of digits $0 - 9$. Images are $28 \times 28$ in gray-scale. For this dataset, we do not use any sort of preprocessing or augmentation. Following Li et al. (2016), we use the following architecture:

$$(32C5) - MP2 - (64C5) - MP2 - 512FC - Softmax \tag{9}$$

Where $C5$ is a $5 \times 5$ ReLU convolution layer, $MP2$ is a $2 \times 2$ max-pooling layer, and $FC$ is a fully connected layer. We adopt Batch Normalization (Ioffe & Szegedy, 2015) into the architecture after every convolution layer. The loss is minimized with Adam. We use dropout (Srivastava et al., 2014)

Table 1: Validation error rates on MNIST and CIFAR-10 datasets, in a binary and ternary setting.

|  | **MNIST** | **CIFAR-10** |
|---|---|---|
| BinaryConnect | $1.29 \pm 0.08\%$ | $9.9\%$ |
| BNN | $1.4\%$ | $10.15\%$ |
| Binary Weight Network[1] | $0.95\%$ | $9.82\%$ |
| LR-net, *binary* (Ours) | **0.53%** | **6.82%** |
| Ternary Weight Network | $0.65\%$ | $7.44\%$ |
| LR-net, *ternary* (Ours) | **0.50%** | **6.74%** |
| Full precision reference | $0.52\%$ | $6.63\%$ |

with a drop rate of $0.5$. Weight decay parameter (on the last full precision layer) is set to $1e-4$. We use a batch size of 256, initial learning rate is 0.01 and is divided by 10 after 100 epochs of training. For the *binary* setting, beta parameter is set to $1e-6$. For the *ternary* setting, probability decay parameter is set to $1e-11$. We report the test error rate after 190 training epochs. The results are presented in Table 1.

## 5.2 CIFAR-10

CIFAR-10 is an image classification benchmark dataset (Krizhevsky, 2009), containing 50K training images and 10K test images from 10 classes. Images are $32 \times 32$ in RGB space. Each image is preprocessed by subtracting its mean and dividing by its standard deviation. During training, we pad each side of the image with 4 pixels, and a random $32 \times 32$ crop is sampled from the padded image. Images are randomly flipped horizontally. At test time, we evaluate the single original $32 \times 32$ image without any padding or multiple cropping. As in Li et al. (2016), we use the VGG inspired architecture:

$$(2 \times 128C3) - MP2 - (2 \times 256C3) - MP2 - (2 \times 512C3) - MP2 - 1024FC - Softmax \quad (10)$$

Where $C3$ is a $3 \times 3$ ReLU convolution layer, $MP2$ is a $2 \times 2$ max-pooling layer, and $FC$ is a fully connected layer. It is worth noting that Courbariaux et al. (2015) and Hubara et al. (2016) use a very similar architecture with two differences, namely adopting an extra fully connected layer and using an L2-SVM output layer, instead of softmax as in our experiments. We adopt Batch Normalization into the architecture after every convolution layer. The loss is minimized with Adam. We use dropout with a drop rate of $0.5$. Weight decay parameter is set to $1e-4$. We use a batch size of 256, initial learning rate is 0.01 and is divided by 10 after 170 epochs of training. For the *binary* setting, beta parameter is set to $1e-6$. For the *ternary* setting, probability decay parameter is set to $1e-11$. We report the test error rate after 290 training epochs. The results are presented in Table 1.

## 5.3 IMAGENET

ImageNet 2012 (ILSVRC2012) is a large scale image classification dataset (Deng et al., 2009), consisting of 1.28 million training images, and 50K validation images from 1000 classes. We adopt the proposed ResNet-18 (He et al., 2016), as in Rastegari et al. (2016) and Li et al. (2016). Each image is preprocessed by subtracting the mean pixel of the whole training set and dividing by the standard deviation. Images are resized so that the shorter side is set to 256. During training, the network is fed with a random $224 \times 224$ crop and images are randomly flipped horizontally. At test time, the center $224 \times 224$ crop is taken. We evaluate only with the single center crop.

The loss is minimized with Adam. Weight decay parameter is set to $1e-5$, we use a batch size of 256 and initial learning rate is 0.01. For the *binary* setting, we found that the beta regularizer is not needed and got the best results when beta parameter is set to $0$. Learning rate is divided by 10 after

---

[1]MNIST and CIFAR-10 results for BWN are taken from Li et al. (2016) where they are marked as BPWNs

Table 2: Validation error rates on ImageNet, in a binary and ternary setting.

|  | ImageNet (top-5) | ImageNet (top-1) |
|---|---|---|
| Binary-Weight-Network | **17%** | **39.2%** |
| XNOR-Net | 26.8% | 48.8% |
| LR-net, *binary* (Ours) | 17.7% | 40.1% |
| Ternary Weight Network | 15.8% | 38.2% |
| LR-net, *ternary* (Ours) | **15.2%** | **36.5%** |
| Full precision reference | 10.76% | 30.43% |

50 and 60 epochs and we report the test error rate after 65 training epochs. For the *ternary* setting, probability decay parameter is set to $1e-12$. For this setting, learning rate is divided by 10 after 30 and 44 epochs and we report the test error rate after 55 training epochs. The results are presented in Table 2.

## 5.4 KERNEL VISUALIZATION

We visualize the first 25 kernels from the first layer of the network trained on MNIST. Fig. 3a, 3b and 3c shows these kernels for the binary, ternary and full precision networks respectively. We can see that our suggested initialization from full precision networks takes advantage of those trained weights and preserves the structure of the kernels.

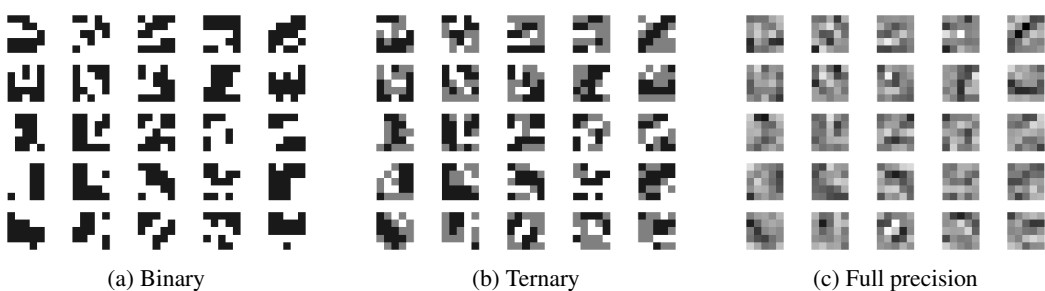

(a) Binary          (b) Ternary          (c) Full precision

Figure 3: Visualization of the first 25 kernels from the first layer of the network trained on MNIST, from the binary, ternary and full precision networks. For the binary and ternary networks, gray, black and white represent 0, -1 and +1, respectively.

## 5.5 BINARY VS TERNARY

In our experiments we evaluate binary and ternary versions of similar networks on similar datasets. When trying to compare the two, it is important to note that the binary results on MNIST and CIFAR-10 are a bit misleading. While the final accuracies are similar, the ternary network work well from the start while the binary networks required more work and careful parameter tuning to achieve similar performance. On harder tasks, like ImageNet classification, the performance gap is significant even when the ternary network discretizes the first layer while the binary did not.

We believe that ternary networks are a much more reasonable model as they do not force each neuron to affect *all* next-layer neurons, and allow us to model low-mean low-variance stochastic weights. The difference in computation time is small as ternary weights do not require multiplications, giving what we consider a worthwhile trade-off between performance and runtime.

## 6   CONCLUSION

In this work we presented a simple, novel and effective algorithm to train neural networks with discrete weights. We showed results on various image classification datasets and reached state-of-the-art results in both the binary and ternary settings on most datasets, paving the way for easier and more efficient training and inference of efficient low-power consuming neural networks.

Moreover, comparing binary and ternary networks we advocate further research into ternary weights as a much more reasonable model than binary weights, with a modest computation and memory overhead. Further work into sparse ternary networks might help reduce, or even eliminate, this overhead compared to binary networks.

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

## A    COMPARISON TO GUMBEL-SOFTMAX

We compare our continuous relaxation to a standard continous relaxation using Gumbel-softmax approximation (Jang et al., 2017). We ran both experiments on CIFAR-10 using the exact same parameters and initialization we use in Sec. 5, for 10 epochs. For the Gumbel-softmax we use a temperature $\tau = 0.1$. We also tried other $\tau$ parameters in the $[0.01, 0.5]$ range and did not find anything that worked noticeably better. We also did not use annealing of the temperature as this is only 10 epochs. Comparing the results in Fig. 4, it is clear that our proposed method works much better then simply using Gumbel-softmax on the weights.

The reason our method works better is twofold. First, we enjoy the lower variance of the local reparamatrization trick since there is one shared source of randomness per pre-activation. Second, the Gumbel-softmax has a trade-off: when the temperature goes down the approximation gets better but the activation saturates and the gradients vanish so it is harder to train. We do not have this problem as we get good approximation with non-vanishing gradients.

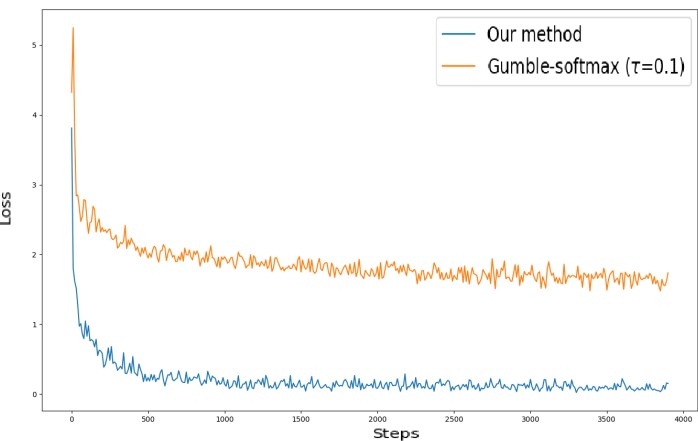

Figure 4: Comparison of CIFAR-10 training with our method and Gumbel-softmax.

# B    AVERAGE ENTROPY

In order to ensure the correctness of our proposed regularization in a ternary setting, we compare the average entropy of the learned probabilities throughout the training, with and without the probability decay term. We examine the average entropy during the training procedure of a network trained on CIFAR-10 in a ternary setting. For the experiment with probability decay, we set the probability decay parameter to $1e{-}11$, as in Sec. 5.

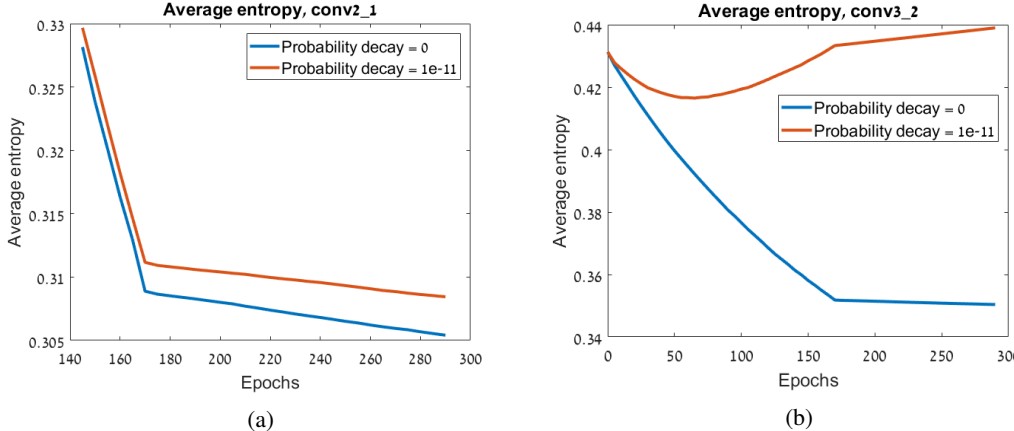

Figure 5: (a) Average entropy of the third convolutional layer, during the last part of the training. (b) Average entropy of the last convolutional layer. The network is trained on CIFAR-10 in a ternary setting.

Fig. 5a shows the average entropy of the third convolutional layer, during the later parts of the training. We can see that the proposed regularization causes a slight increase in entropy, which is what we aimed to achieve. Note that the 'kink' in the graph is due to lowering the learning rate after 170 epochs. For some of the layers this regularization has a much more significant effect. Fig. 5b shows the average entropy throughout the training for the last convolutional layer. For this layer, the entropy at the end of the training is higher than the entropy in initialization. Looking at the probabilities themselves helps better explain this behavior.

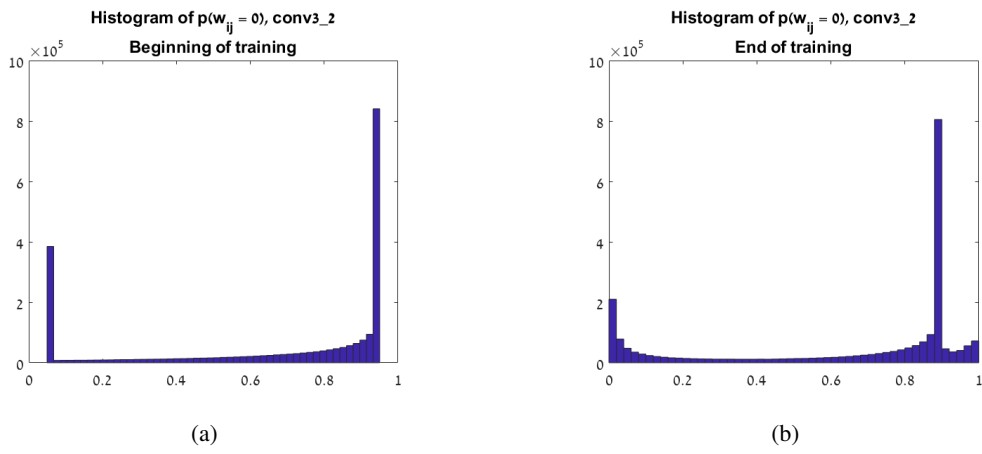

Figure 6: Histogram of $p(w_{ij} = 0)$ for the last convolutional layer. (a) Initialized value, beginning of training. (b) Learned values, end of training. The network is trained on CIFAR-10 in a ternary setting.

Fig 6a shows a histogram of the initialized probability $p(w_{ij} = 0)$ of this layer. Fig 6b shows a histogram of these learned probabilities at the end of the training. It can be seen that for this

layer there were many weights that were assigned a very high (or very low) probability value for $p(w_{ij} = 0)$. However, the probability decay regularization pushed the learned values to be less deterministic, causing the increase in entropy observed in Fig. 5b.

