# OpenReview forum: "Learning Discrete Weights Using the Local Reparameterization Trick"
_ICLR.cc/2018/Conference — Accept (Poster)_

### Official Review · AnonReviewer2 · 2017-11-24
**The idea of making use of the local parameterisation trick to learn discrete networks is straight forward but novel and leads to SOTA results.**

**Rating:** 7
**Confidence:** 3

**Review:**

Summary of the paper:
The paper suggests to use stochastic parameters in combination with the local reparametrisation trick (previously introduced by Kingma et al. (2015)) to train neural networks with binary or ternary wights. Results on MNIST, CIFAR-10 and ImageNet are very competitive.

Pros:
- The proposed method leads to state of the art results .
- The paper is easy to follow and clearly describes the implementation details needed to reach the results.

Cons:
- The local reprarametrisation trick it self is not new and applying it to a multinomial distribution (with one repetition) instead of a Gaussian is straight forward, but its application for learning discrete networks is to my best knowledge novel and interesting.

It could be nice to include the results of Zuh et al (2017) in the results table and to indicate the variance for different samples of weights resulting from your methods in brackets.


Minor comments:
- Some citations have a strange format: e.g. “in Hubara et al. (2016); Restegari et al. (2016)“ would be better readable as   “by Hubara et al. (2016) and Restegari et al. (2016)“
-  To improve notation, it could be directly written that W is the set of all w^l_{i,j} and \mathcal{W} is the joint distribution resulting from independently sampling from  \mathcal{W}^l_{i,j}.
- page 6: “on the last full precision network”: should probably be “on the last full precision layer”
                    “ distributions has” ->  “ distributions have”

---

### Official Review · AnonReviewer1 · 2017-11-27
**Simple idea that seem to work but the novelty is limited and some regularization choices might not do what is expected.**

**Rating:** 6
**Confidence:** 4

**Review:**

This paper proposes training binary and ternary weight distribution networks through the local reparametrization trick and continuous optimization. The argument is that due to the central limit theorem (CLT) the distribution on the neuron pre-activations is approximately Gaussian, with a mean given by the inner product between the input and the mean of the weight distribution and a variance given by the inner product between the squared input and the variance of the weight distribution. As a result, the parameters of the underlying discrete distribution can be optimized via backpropagation by sampling the neuron pre-activations with the reparametrization trick. The authors further propose appropriate initialisation schemes and regularization techniques to either prevent the violation of the CLT or to prevent underfitting. The method is evaluated on multiple experiments.

This paper proposed a relatively simple idea for training networks with discrete weights that seems to work in practice. My main issue is that while the authors argue about novelty, the first application of CLT for sampling neuron pre-activations at neural networks with discrete r.v.s is performed at [1]. While [1] was only interested in faster convergence and not on optimization of the parameters of the underlying distribution, the extension was very straightforward. I would thus suggest that the authors update the paper accordingly.

Other than that, I have some other comments:
- The L2 regularization on the distribution parameters for the ternary weights is a bit ad-hoc; why not penalise according to the entropy of the distribution which is exactly what you are trying to achieve?
- For the binary setting you mentioned that you had to reduce the entropy thus added a “beta density regulariser”. Did you add R(p) or log R(p) to the objective function? Also, with alpha, beta = 2 the beta density is unimodal with a peak at p=0.5; essentially this will force the probabilities to be close to 0.5, i.e. exactly what you are trying to avoid. To force the probability near the endpoints you have to use alpha, beta < 1 which results into a “bowl” shaped Beta distribution. I thus wonder whether any gains you observed from this regulariser are just an artifact of optimization.
- I think that a baseline (at least for the binary case) where you learn the weights with a continuous relaxation, such as the concrete distribution, and not via CLT would be helpful. Maybe for the network to properly converge the entropy for some of the weights needs to become small (hence break the CLT).

[1] Wang & Manning, Fast Dropout Training.

Edit: After the authors rebuttal I have increased the rating of the paper:
- I still believe that the connection to [1] is stronger than what the authors allude to; eg. the first two paragraphs of sec. 3.2 could easily be attributed to [1].
- The argument for the entropy was to include a term (- lambda * H(p)) in the objective function with H(p) being the entropy of the distribution p. The lambda term would then serve as an indicator to how much entropy is necessary.
- There indeed was a misunderstanding with the usage of the R(p) regularizer at the objective function (which is now resolved).
- The authors showed benefits compared to a continuous relaxation baseline.

---

### Official Review · AnonReviewer3 · 2017-11-28
**The Kingma's reparametrization trick for binary and ternary nets**

**Rating:** 6
**Confidence:** 3

**Review:**

This paper introduces the LR-Net, which uses the reparametrization trick inspired by a similar component in VAE. Although the idea of reparametrization itself is not new, applying that for the purpose of training a binary or ternary network, and sample the pre-activations instead of weights is novel.  From the experiments, we can see that the proposed method is effective.

It seems that there could be more things to show in the experiments part. For example, since it is using a multinomial distribution for weights, it makes sense to see the entropy w.r.t. training epochs. Also, since the reparametrization is based on the Lyapunov Central Limit Theorem, which assumes statistical independence, a visualization of at least the correlation between the pre-activation of each layer would be more informative than showing the histogram.

Also, in the literature of low precision networks, people are concerning both training time and test time computation demand. Since you are sampling the pre-activations instead of weights, I guess this approach is also able to reduce training time complexity by an order. Thus a calculation of train/test time computation could highlight the advantage of this approach more boldly.

---

### Decision · Program_Chairs · 2018-01-29
**ICLR 2018 Conference Acceptance Decision**

**Decision:**

Accept (Poster)

**Comment:**

Well written paper on a novel application of the local reprarametrisation trick to learn networks with discrete weights. The approach achieves state-of-the-art results.

Note: I apreciate that the authors added a comparison to the Gumbel-softmax continuous relaxation approach during the review period, following the suggestion of a reviewer. This additional comparison strengthens the paper.